# Hot Electron Extraction in SWCNT/TiO_2_ for Photocatalytic H_2_ Evolution from Water

**DOI:** 10.3390/nano12213826

**Published:** 2022-10-29

**Authors:** Masahiro Yamagami, Tomoyuki Tajima, Zihao Zhang, Jun Kano, Ki-ichi Yashima, Takana Matsubayashi, Huyen Khanh Nguyen, Naoto Nishiyama, Tomoya Hayashi, Yutaka Takaguchi

**Affiliations:** 1Graduate School of Environmental and Life Science, Okayama University, 3-1-1 Tsushima-Naka, Kita-ku, Okayama 700-8530, Japan; 2Graduate School of Natural Science and Technology, Okayama University, 3-1-1 Tsushima-naka, Kita-ku, Okayama 700-8530, Japan; 3Department of Material Design and Engineering, Faculty of Sustainable Design, University of Toyama, Toyama 930-8555, Japan

**Keywords:** single-walled carbon nanotube, photocatalyst, hydrogen evolution, water splitting, hot electron extraction

## Abstract

Single-walled carbon nanotube (SWCNT)/TiO_2_ hybrids were synthesized using 1,10-bis(decyloxy)decane-core PAMAM dendrimer as a molecular glue. Upon photoirradiation of a water dispersion of SWCNT/TiO_2_ hybrids with visible light (λ > 422 nm), the hydrogen evolution reaction proceeded at a rate of 0.95 mmol/h·g in the presence of a sacrificial agent (1-benzyl-1,4-dihydronicotinamide, BNAH). External quantum yields (EQYs) of the hydrogen production reaction photosensitized by (6,5), (7,5), and (8,3) tubes were estimated to be 5.5%, 3.6%, and 2.2%, respectively, using monochromatic lights corresponding to their E_22_ absorptions (570 nm, 650 nm, and 680 nm). This order of EQYs (i.e., (6,5) > (7,5) > (8,3)SWCNTs) exhibited the dependence on the C_2_ energy level of SWCNT for EQY and proved the hot electron extraction pathway.

## 1. Introduction

Single-walled carbon nanotubes (SWCNTs) have attracted attention for their application as a photoelectric conversion material due to their outstanding solar light absorption property [1]. The optical absorption of semiconducting SWCNTs reveals sets of chirality-dependent absorption bands in the near-infrared and visible wavelength regions, which are labeled the first (E_11_) and second (E_22_) transitions, corresponding to the discrete energetic transitions of one-dimensional van Hove singularities [2]. The energy level of the second excited state (E_2_ state) is higher than that of the first excited state (E_1_ state). Hence, hot electron extraction directly from the E_2_ state is effective to improve the performance of photovoltaics and photocatalysts based on SWCNT-light absorbers. However, regarding a SWCNT/C_60_ heterojunction, which is often used in organic solar cells, the relaxation from the E_2_ state to E_1_ or ground states suppresses the hot electron extraction from SWCNT to C_60_ [3]. As a result, the internal quantum efficiency (IQE) of SWCNT/C_60_ solar cells depends on an energetic offset between the lowest molecular orbital (LUMO: C_1_), corresponding to the E_1_ state, of the nanotube and that of C_60_ [4]. In other words, the IQE of SWCNT/C_60_ solar cells is not affected by the energy level of the E_2_ state (LUMO+1: C_2_) of SWCNTs, even upon E_22_ photoexcitation with visible light. Recently, we developed SWCNT/C_60_ photocatalysts that act as hydrogen evolution photocatalysts [5,6,7,8,9]. As in the case of SWCNT/C_60_ solar cells, the external quantum yield (EQY) of the photocatalytic hydrogen production increased in the order (7,5)SWCNT (0.17%) < (6,5)SWCNT (0.35%) < (8,3)SWCNT (1.5%) with increasing LUMO energy levels of the SWCNTs, despite photoexcitation using E_22_ absorption to generate the E_2_ state [8]. Density functional theory (DFT) calculation has shown that E_22_ excitation does not induce electron injection to C_60_ in the (6,5)SWCNT/C_60_ interface, although, with E_11_ excitation, ultrafast electron transfer (τ < 200 fs) takes place from (6,5)SWCNT to C_60_ [10]. These observations indicate that C_60_ is not capable of extracting hot electrons from SWCNTs. On the other hand, Parkinson and co-workers fabricated SWCNT heterojunctions with atomically flat surfaces of TiO_2_ and SnO_2_, where higher-energy second excitonic SWCNT transitions produce more photocurrent [11]. Because of the continuum of states within the metal-oxide conduction band with a density that increases with increasing energy above the conduction band minimum, rates of carrier injection from E_2_ of SWCNT to TiO_2_ or SnO_2_ are competitive with fast hot-exciton relaxation processes. In this context, the construction of similar photocatalytic systems is of interest in order to make photocatalytic reactions using SWCNTs more efficient. In this paper, we synthesized SWCNT/TiO_2_ nanohybrids to demonstrate their photocatalytic activity for hydrogen evolution from water through the hot electron extraction from the E_2_ state of SWCNT to TiO_2_.

## 2. Materials and Methods

### 2.1. Reagents and Chemicals

SWCNTs were purchased from Sigma-Aldrich Co (St. Louis, MO, USA). TiO_2_ was synthesized according to a previous report [12]. BDD-dendrimer(COOH) [13] and SWCNT/BDD-dendrimer(COOH) [14] were prepared according to our previous reports. All other reagents were purchased from Kanto Kagaku Co., Ltd. (Tokyo, Japan), Aldrich Chemical Co. (St. Louis, MO, USA), and Tokyo Kasei Co., Ltd. (Tokyo, Japan). All chemicals were used as received. 

### 2.2. Characterization

The absorption data were recorded on a UV-3150 spectrophotometer (Shimadzu, Tokyo, Japan) using a standard cell with a path length of 10 mm. SEM measurements for the composites were conducted using a JEM-2100 (JEOL Ltd., Tokyo, Japan). Specimens for the measurements were prepared by applying a few drops of the sample solution onto a dedicated grid, and then evaporating the solvent. Raman spectra were obtained on a JASCO NRS-5100 (JASCO Co., Japan) using laser excitation at 532 nm. 

### 2.3. Synthesis of SWCNT/BDD-Dendrimer(COOH) Nanohybrids

SWCNT/BDD-dendrimer(COOH) nanohybrids were synthesized as follows in accordance with the literature [14]. SWCNTs (1.0 mg) were added to a solution of BDD-dendrimer(COOH) (1.0 mg, 0.13 mM) in H_2_O (5.0 mL) and then sonicated using a bath-type ultrasonifier (AS ONE Vs-D100, 24 kHz/31 kHz, 110 W) at a temperature below 25 °C for 4 h. The centrifugation (3000 G) of the suspension for 30 min gave a stock solution of SWCNT/BDD-dendrimer(COOH) nanohybrids as a black-colored transparent supernatant.

### 2.4. Synthesis of SWCNT/TiO_2_/Pt Hybrids

Here, 1.0 wt% Pt-loaded TiO_2_ particles were prepared using the previously reported method [12]. The solid of TiO_2_ (25 mg) was added to distilled water (22.5 mL), H_2_PtCl_6_ (0.66 mg, 1.28 μmol), and methanol (2.5 mL), and then irradiated by a 300 W Xe lamp (300 W, MAX-303, Asahi Spectra Co., M.C., without an optical filter) with stirring for 4 h at room temperature. After the irradiation, the gray solid was rinsed with methanol and dried under a vacuum. The SWCNTs were adsorbed onto TiO_2_ by stirring the TiO_2_/Pt (10 mg) in a mixture of SWCNT/BDD-dendrimer(COOH) nanohybrids (125 μL, SWCNT content 0.025 mg) in water (10 mL) at room temperature for 30 min and immersed overnight in the dark. After that, the supernatant was removed by decantation and the sample was dried and kept in the dark (denoted as SWCNT/TiO_2_/Pt).

### 2.5. Photocatalytic Hydrogen Evolution Using SWCNT/TiO_2_/Pt Hybrids

SWCNT/TiO_2_/Pt (10 mg) and BNAH (38.6 mg, 180 μmol) were dissolved in deionized water (150 mL) in a Pyrex reactor. Upon vigorous stirring, the solution was irradiated with a 300 W Xenon arc light (300 W, MAX-303, Asahi Spectra Co., M.C., Tokyo, Japan) through the cut-off filter (λ > 422 ± 5 nm: Asahi Spectra Co., M.C., 25 φ) and bandpass filter (λ = 570 nm, 650 nm, 680 nm: Asahi Spectra Co., M.C., 25 φ). After a designated period of time, the cell containing the reaction mixture was connected to a gas chromatograph (Shimadzu, TCD, molecular sieve 5 A: 2.0 m × 3.0 mm, Ar carrier gas) to measure the amount of H_2_ above the solution (Appendix A).

## 3. Results and Discussion

Water-dispersible SWCNT/BDD-dendrimer nanohybrids were synthesized by the physical modification of SWCNTs with poly(amidoamine) dendrimer with a 1,10-bis(decyloxy)decane core and carboxy (–COOH) terminals, BDD-dendrimer(COOH) (Figure 1a) [12,13]. Pt-loaded TiO_2_ mesocrystals (TiO_2_/Pt) (25 mg) were prepared by a conventional photochemical deposition method [12]. The Pt loading on TiO_2_ was confirmed by the hydrogen production activity (0.71 μmol/h·mg) under UV irradiation. To a water dispersion (10 mL) of TiO_2_/Pt (10 mg), a solution of SWCNT/BDD-dendrimer(COOH) nanohybrids (125 μL, SWCNT content 0.025 mg) was added. The mixture was stirred for 30 min and immersed overnight in the dark. The solvent was removed by decantation to obtain SWCNT/TiO_2_/Pt (Figure 1b). The amount of SWCNTs adsorbed on the TiO_2_ surface was estimated to be 21 µg per 10 mg of TiO_2_/Pt using the absorption spectrum of supernatant solution after the hybridization of SWCNTs with TiO_2_ (Appendix A).

Figure 2 shows SEM images of TiO_2_ and SWCNTs/TiO_2_/Pt. The TiO_2_ mesocrystals were plate-like structures with a particle size of 10 μm (Figure 2a), as previously reported [12]. The plate-like structure was retained after the attachment of SWCNT/BDD-dendrimer(COOH) nanohybrids to TiO_2_ (Figure 2b). TiO_2_ particles on the plate mesocrystals were stripped off by ultrasonic treatment during the Pt loading or SWCNTs attachment process. Although the amount of Pt and SWCNTs on the surface of the TiO_2_ crystals was so small that they could not be observed by energy dispersive X-ray spectroscopy (EDX), HR-SEM images show the Pt nanoparticles and nanofibers on the TiO_2_ mesocrystals (Appendix A). The SWCNTs/TiO_2_/Pt exhibited a Brunauer–Emmett–Teller (BET) surface area of 54.7 m^2^ g^−1^.

The formation of SWCNT/TiO_2_/Pt nanohybrids was confirmed by absorption, 2D excitation/emission, and Raman spectra. The absorption spectra of SWCNT/TiO_2_/Pt (blue line) exhibit the characteristic absorption bands derived from SWCNTs that appeared at 400–700 nm (Figure 3). The absorption originating from (6,5)SWCNT (λ_max_ 570 nm) along with a small shoulder at around 670 nm originating from (8,3) and (7,5)SWCNTs were observed, almost the same as that of SWCNT/BDD-dendrimer(COOH) (orange line). Two-dimensional excitation/emission spectra show the quenching of the fluorescence of SWCNTs after the hybridization with TiO_2_/Pt due to photoinduced electron transfer from SWCNTs to TiO_2_ (Appendix A).

Figure 4 shows the Raman spectrum of the SWCNT/TiO_2_/Pt hybrids. Raman shifts for the G band (1585 cm^−1^), D band (1316 cm^−1^), and G’ band (2622 cm^−1^) of the SWCNTs were observed, where the G/D ratio (3.30) did not change before or after the attachment of SWCNT onto TiO_2_, indicating that the sp^2^ carbon of SWCNTs was not changed to sp^3^ carbon. Peaks originating from the anatase crystal of TiO_2_ were observed at 403 cm^−1^ (B_1g_), 517 cm^−1^ (A_1g_), 156 cm^−1^, and 643 cm^−1^ (E_g_) [15]. These observations indicate that SWCNTs are adsorbed on the surface of TiO_2_.

Parkinson and co-workers reported that hot electron injection from the SWCNT E_22_ state to TiO_2_ is more efficient than electron injection from the relaxed SWCNT E_11_ state at the SWCNT/TiO_2_ heterojunction due to a higher density of states (DOS) at E_22_ than at E_11_ of SWCNTs and the continuum of states within the TiO_2_ conduction band with a density that increases with increasing energy above the conduction band edge [11]. In marked contrast with SWCNT/C_60_ heterojunctions [3,10], hot electron extraction from the SWCNT E_22_ state (C_2_) is fast enough to compete with relaxation to the E_11_ state (C_1_) so that the relative photon conversion efficiency (RPCE) upon E_22_ photoexcitation is higher than that upon E_11_ photoexcitation. In this context, we expected photocatalytic hydrogen evolution from water using SWCNT/TiO_2_/Pt nanohybrids via direct electron extraction from SWCNT E_22_ states (C_2_), of which the energy level diagram is shown in Figure 5. Higher-energy second excitonic SWCNT transition under visible-light irradiation leads to hot electron extraction from the SWCNT E_22_ state (C_2_) to the TiO_2_ conduction band, followed by the electron migration to the Pt co-catalyst to induce the hydrogen evolution reaction. The remaining hole in the SWCNT valence bands (V_2_) is consumed by simultaneous hole migration to a sacrificial donor molecule, 1-benzyl-1,4-dihydronicotinamide (BNAH). The efficiency of the hydrogen evolution reaction is dominated by the efficiency of the hot electron extraction from SWCNT E_22_ to TiO_2_, because the electron injection from the SWCNT E_11_ state (C_1_) to TiO_2_ is relatively slow due to the small driving forces, although the hot-electron injection rate from C_2_ to TiO_2_ is competitive with hot-exciton relaxation processes, as described by Parkinson et al. [11].

Figure 6 shows the time course of the photocatalytic hydrogen evolution reaction over SWCNT/TiO_2_/Pt under visible-light irradiation (λ > 422 nm). The hydrogen production rate of 9.5 μmol/h was observed (Figure 6, ●). The hydrogen evolution reaction continued until all of the sacrificial agent, BNAH, was consumed, and there was no induction period (Appendix A). In contrast, no production of hydrogen was detected using TiO_2_/Pt without SWCNTs under the same conditions (Figure 6, ■), indicating that the SWCNTs act as photosensitizers and the hydrogen production reaction proceeds via electron extraction from SWCNTs on the surface of TiO_2_. To compare the electron-extracting ability of TiO_2_, commercially available P25 was used to synthesize SWCNT/TiO_2_(P25)/Pt. Under the same reaction conditions, the hydrogen production rate of SWCNT/TiO_2_(P25)/Pt was 7.3 µmol/h (Appendix A), which is less active than that of SWCNT/TiO_2_(mesocrystal)/Pt (9.5 µmol/h). The higher activity with TiO_2_ mesocrystals may be due to the suppression of charge recombination at the SWCNT/TiO_2_ interface. A similar result using black phosphorous/TiO_2_ (mesocrystal) interface was described by Fujitsuka and co-workers [16].

To obtain insight into this free-carrier generation process in the SWCNT/TiO_2_ heterojunction, we compared the chirality dependence of EQY of the hydrogen evolution reaction using the SWCNT/TiO_2_ heterojunction with that using SWCNT/C_60_ upon E_22_ photoexcitation of SWCNTs. In our previous reports [8,9], we found a commensurate reduction of EQY in the offset of the energy levels (driving force) between SWCNT C_1_ and C_60_ LUMO (Figure 7). (8,3)SWCNT shows the highest EQY among (6,5), (7,5), and (8,3)SWCNTs because of the electron transfer from SWCNT to C_60_ after the relaxation from the SWCNT E_22_ state to the SWCNT E_11_ state, as in the case of SWCNT/C_60_ solar cells. If the hot electron extraction from the SWCNT E_22_ state to C_60_ had occurred, the EQY would depend on the energy levels of SWCNT C_2_, i.e., (6,5)SWCNT would represent the highest EQY, and (8,3)SWCNT would show the lowest EQY.

In this context, we investigated the photocatalytic activity of SWCNT/TiO_2_/Pt upon chirality-selective photoexcitation using monochromatic light irradiation at 570, 650, and 680 nm, which are the E_22_ absorptions of (6,5), (7,5), and (8,3)SWCNTs, respectively. In a typical experiment, 150 mL of an aqueous dispersion of SWCNT/TiO_2_/Pt (10 mg) and 1-benzyl-4-dihydronicotinamide (BNAH; 38.6 mg, 180 μmol/h) was exposed to monochromatic light (570, 650, or 680 nm) using a 300 W Xenon arc lamp with bandpass filters while being stirred vigorously at 25 °C. After the designated period, the gas phase above the solution was analyzed by gas chromatography. Figure 8a (●) shows plots of the total amount of H_2_ produced versus time using monochromatic light irradiation at 570 nm. A steady generation of H_2_ (2.2 μmol/h) was observed without an induction period or a decrease in activity during 3 h of irradiation. Compared with the H_2_ generated by the use of monochromatic light irradiation at 650 or 680 nm, 1.7 µmol/h (Figure 8a (■)) and 1.3 µmol/h (Figure 8a (◆)), respectively, the amount of H_2_ evolution under 570 nm irradiation was the highest (2.2 μmol/h, Figure 8a (●)). The EQYs were in the same order as for the hydrogen production rate: 5.5% for (6,5)SWCNT > 3.6% for (7,5)SWCNT > 2.2% for (8,3)SWCNT. Notably, this order of EQYs is consistent with the energy levels of the second excitonic state (C_2_) for (6,5), (7,5), and (8,3)SWCNTs, –3.30, –3.54, and –3.62 eV, respectively (Figure 9), and is the same as previous reports on the relative photon conversion efficiency (RPCE) of the SWCNT/TiO_2_ heterojunction. This result indicated that the hot electron injection from the second excitonic state of SWCNTs to TiO_2_ leads to a hydrogen evolution reaction in marked contrast to CNT photocatalyst based on the SWCNT/C_60_ heterojunction, where the electron extraction from SWCNT to C_60_ occurred after the inter-band transition from the E_22_ state (C_2_) to the E_11_ state (C_1_). Furthermore, the SWCNT/TiO_2_/Pt photocatalyst exhibited higher EQYs than the previously reported SWCNT/C_60_/Pt(II) photocatalyst. For example, upon 570 nm photoirradiation (E_22_ absorption of (6,5)SWCNT), the EQY of SWCNT/TiO_2_/Pt, 5.5%, is 16 times higher than that of SWCNT/C_60_/Pt(II), 0.35%.

## 4. Conclusions

In summary, we have prepared a visible-light-responsive TiO_2_ photocatalyst, SWCNT/TiO_2_/Pt, by mixing TiO_2_/Pt and SWCNT/BDD-dendrimer nanohybrids. Since BDD-dendrimer can act as a molecular glue that does not suppress the electron transfer between SWCNT and TiO_2_, the photoinduced electron transfer from SWCNT to TiO_2_ proceeds very smoothly to form a charge-separated state (SWCNT^+^/TiO_2_^−^). The dependence on the C_2_ energy level of SWCNT for the EQY of the hydrogen evolution reaction upon E_22_ photoexcitation proved the hot electron extraction pathway. Interestingly, the EQYs are higher than those of the previous reports employing SWCNT/C_60_/Pt(II) as a photocatalyst because of the difference in the charge collection process where electron extraction takes place after the relaxation from the SWCNT E_22_ state to the SWCNT E_11_ state. Further studies on the SWCNT heterojunction with metal oxides to enhance the efficiency of the hot electron extraction pathway for solar hydrogen production are currently in progress in our laboratories.

## Figures and Tables

**Figure 1 nanomaterials-12-03826-f001:**
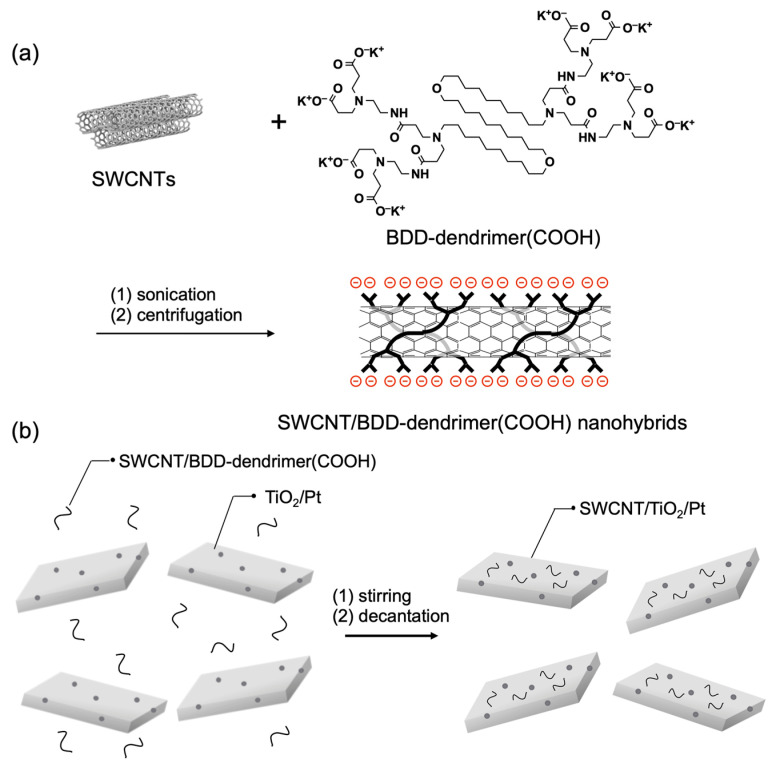
Fabrication of (**a**) the SWCNT/BDD-dendrimer(COOH) nanohybrids, and (**b**) SWCNT/TiO_2_/Pt nanohybrids.

**Figure 2 nanomaterials-12-03826-f002:**
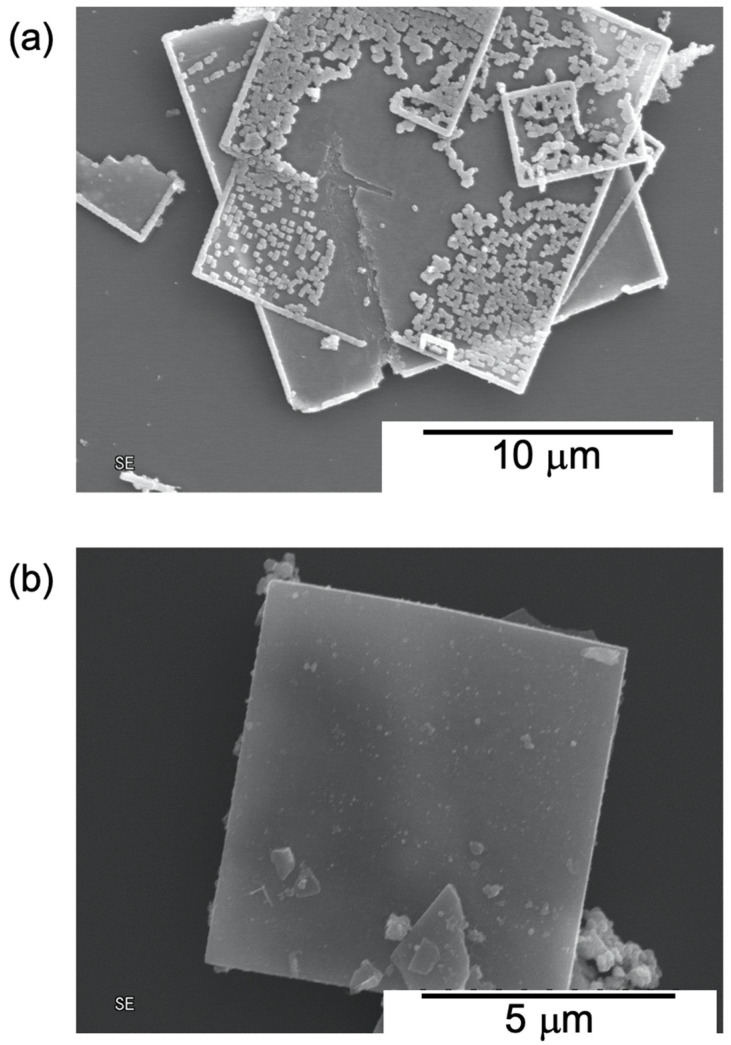
SEM images of (**a**) TiO_2_ and (**b**) SWCNT/TiO_2_/Pt.

**Figure 3 nanomaterials-12-03826-f003:**
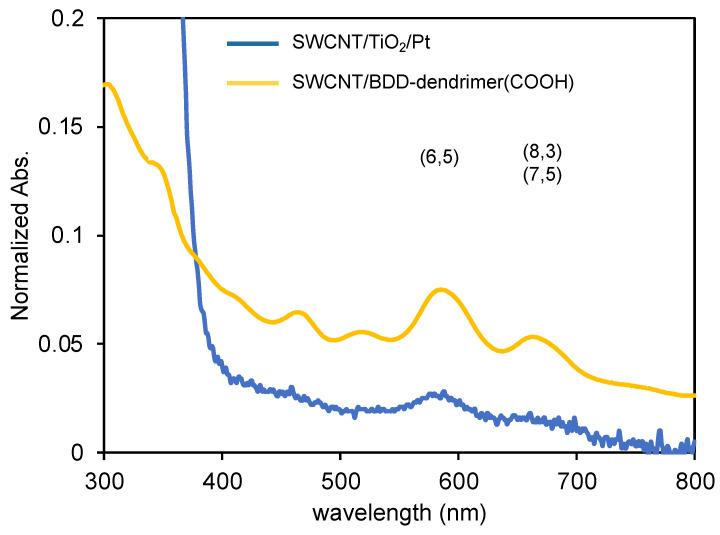
Absorption spectra of SWCNT/TiO_2_/Pt (blue line) and SWCNT/BDD-dendrimer(COOH) nanohybrids (orange line).

**Figure 4 nanomaterials-12-03826-f004:**
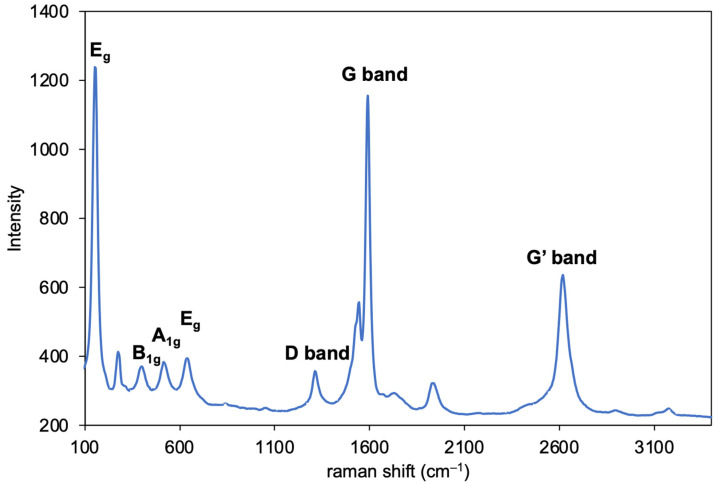
Raman spectrum of SWCNT/TiO_2_/Pt hybrids.

**Figure 5 nanomaterials-12-03826-f005:**
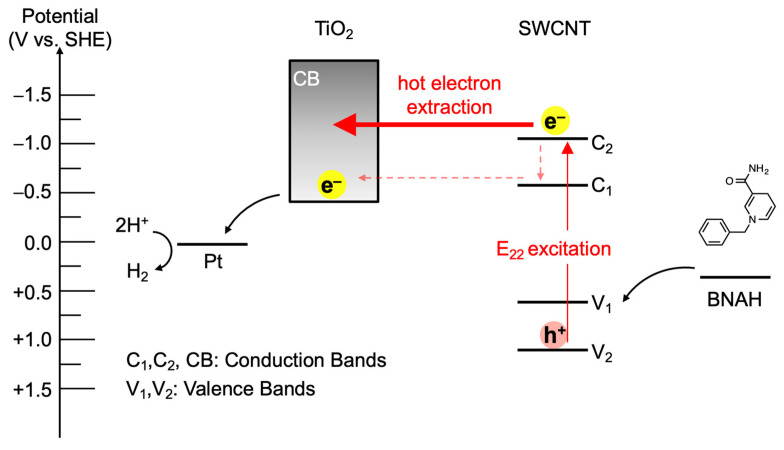
Energy-level diagram of the hydrogen evolution reaction using SWCNT/TiO_2_/Pt as a photocatalyst in the presence of BNAH as a sacrificial donor.

**Figure 6 nanomaterials-12-03826-f006:**
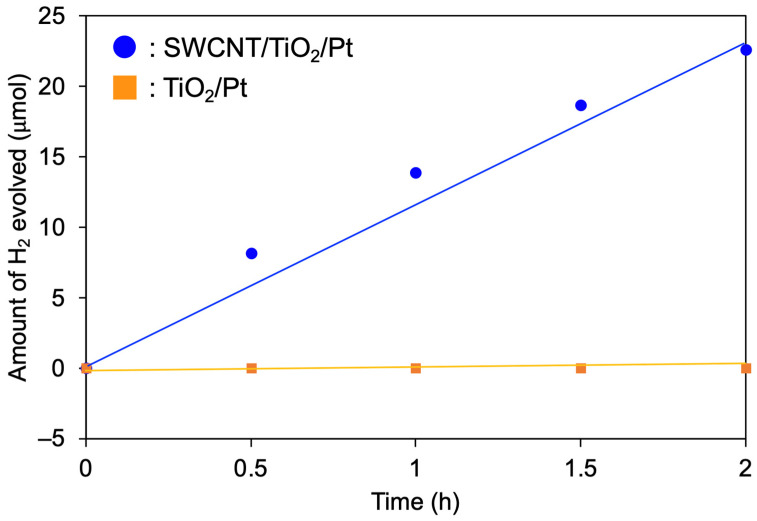
Time course of H_2_ evolution from water over SWCNT/TiO_2_/Pt (blue line) and TiO_2_/Pt (orange line) under visible-light irradiation (λ > 422 nm).

**Figure 7 nanomaterials-12-03826-f007:**
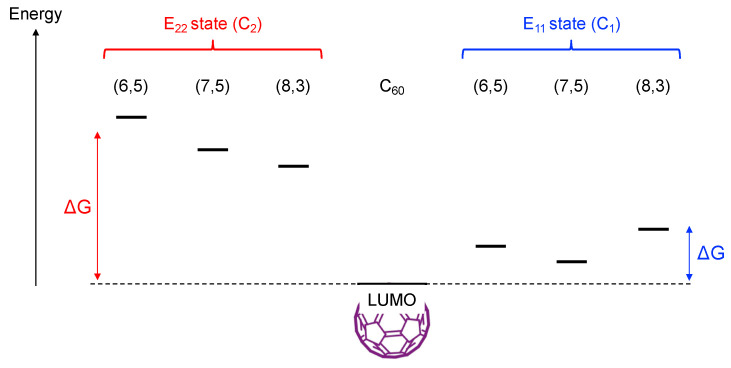
Driving force of electron extraction using SWCNT/C_60_ heterojunction from SWCNT E_22_ state (**left** side) and SWCNT E_11_ state (**right** side).

**Figure 8 nanomaterials-12-03826-f008:**
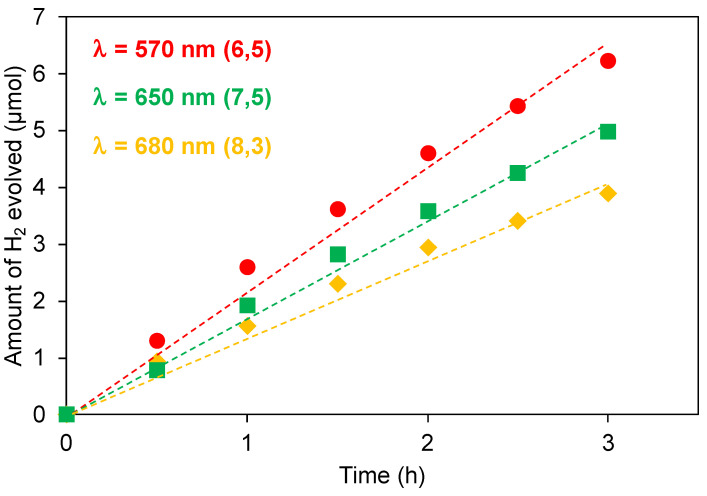
Time course of the H_2_ evolution under irradiation at 570 nm (E_22_ of the (6,5)SWCNT; ●), 650 nm (E_22_ of (7,5)SWCNT; ■), and 680 nm (E_22_ of the (8,3)SWCNT; ▲).

**Figure 9 nanomaterials-12-03826-f009:**
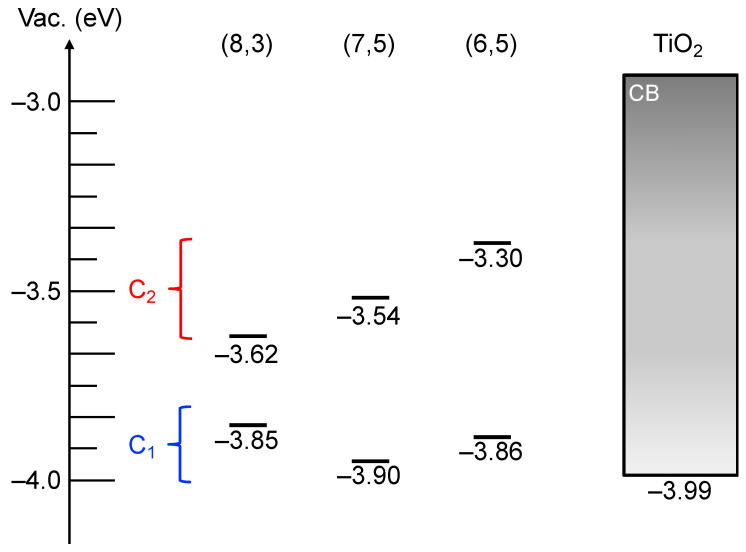
Energy level diagram of SWCNT/TiO_2_ heterojunctions. Gray bar reveals the conduction band of TiO_2_.

## Data Availability

The data presented in this study are available on request from the corresponding author.

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
