# Peer review of "Hot Electron Extraction in SWCNT/TiO2 for Photocatalytic H2 Evolution from Water"

_nanomaterials, 2022, doi:10.3390/nano12213826_

Round 1

Reviewer 1 Report

This manuscript was written by Masahiro Yamagami et al. report on SWCNT/TiO2 hybrids for applying photocatalytic water splitting. From my perspective, this manuscript contains information that can interest the scientific community, and I recommend its publication. However, amendments must be made before the final publication. Below are listed my observations.

1.        Please provides the TEM, HR-TEM, and TEM elemental mapping to analyze the SWCNT/TiO2 hybrids in the revised manuscript or supporting information.

2.        Photocatalytic hydrogen production: the amount of catalyst? The condition? How did you collect the gas and measure it?

3.        In this study, the surface area is a significant parameter. Therefore, I strongly suggest adding the BET surface area measurement for all samples in the revised manuscript or supporting information.

4.        How economical when it comes to production on an industrial level. Did the authors have any estimation?

5.        The photocatalytic water splitting should be compared with the same weight of commercial p25 TiO2.

6.        Some basic characterizations like PL or EPR should be tested to characterize the prepared sample.

Reviewer 2 Report

The authors describe the preparation of a photocatalyst consisting of titania doped with platinum, and sensitized by Single-walled carbon nanotubes (SWCNTs).The photocatalyst is prepared using a traditional procedure consisting of suspending titania nanoparticles in a solution of hexachloroplatinic acid, and subsequent illumination with ultraviolet light with the aim of reducing the Pt absorbed on the surface of titanium oxide. Subsequently, the doped titania is immersed in a suspension of nanotubes functionalized with poly(amidoamine) dendrimer in order to obtain the anchorage of nanotubes to titania through the carboxyl groups of with poly(amidoamine) dendrimer.

The photocatalyst thus prepared is used in the photocatalytic production of hydrogen using 1-benzyl-1,4-dihydronicotinamide  as SED. The authors highlight how the excited state E2,2 is the one that allows a more efficient production of hydrogen due to the high energy level of the excited state that allows to have a greater driving force for the injection of hydrogen into the titania rather than having a recombination (which is more likely in the case of the excited state E1, 1).

The manuscript is clear and the results well summarized. From the point of view of methodology I find some inaccuracies that should be corrected in order to accept this work:

1)      The amount of dendrimer bound to titanium oxide is unknown. If everything is supposed to bond the titania after staining, it would be necessary to make a UV-vis analysis of the supernatant solution and show that there is no more. Otherwise, it is necessary to quantify through analytical measures.

2)      The hydrogen production measurement involves a hydrogen sampling every 30 minutes. This is too long a time to be able to say that there is no induction period that is normally visible in the first 10-15 minutes. In addition, the time in which the reaction is followed is too short (2h). To have information also on the stability of the system would take at least 24 hours.

Round 2

Reviewer 1 Report

The authors have revised the manuscript according to the comments. Therefore, I recommend that the manuscript could be accepted for publication.

Reviewer 2 Report

I appreciate the revisions made by the authors and I believe that the manuscript can now be accepted for publication in Nanomaterials.